# Development of an Inflammation-Triggered In Vitro “Leaky Gut” Model Using Caco-2/HT29-MTX-E12 Combined with Macrophage-like THP-1 Cells or Primary Human-Derived Macrophages

**DOI:** 10.3390/ijms24087427

**Published:** 2023-04-18

**Authors:** Nguyen Phan Khoi Le, Markus Jörg Altenburger, Evelyn Lamy

**Affiliations:** 1Molecular Preventive Medicine, University Medical Center and Faculty of Medicine, University of Freiburg, 79108 Freiburg, Germany; phan.khoi.nguyen.le@uniklinik-freiburg.de; 2Department of Operative Dentistry and Periodontology, University Medical Center and Faculty of Medicine, University of Freiburg, 79108 Freiburg, Germany; markus.altenburger@uniklinik-freiburg.de

**Keywords:** leaky gut, inflammatory bowel disease (IBD), triple-culture, Caco-2, HT29-MTX-E12, THP-1, monocyte-derived macrophage, inflammation, TEER, FITC-dextran 4 kDa (FD4), pro-inflammatory cytokines

## Abstract

The “leaky gut” syndrome describes a damaged (leaky) intestinal mucosa and is considered a serious contributor to numerous chronic diseases. Chronic inflammatory bowel diseases (IBD) are particularly associated with the “leaky gut” syndrome, but also allergies, autoimmune diseases or neurological disorders. We developed a complex in vitro inflammation-triggered triple-culture model using 21-day-differentiated human intestinal Caco-2 epithelial cells and HT29-MTX-E12 mucus-producing goblet cells (90:10 ratio) in close contact with differentiated human macrophage-like THP-1 cells or primary monocyte-derived macrophages from human peripheral blood. Upon an inflammatory stimulus, the characteristics of a “leaky gut” became evident: a significant loss of intestinal cell integrity in terms of decreased transepithelial/transendothelial electrical resistance (TEER), as well as a loss of tight junction proteins. The cell permeability for FITC-dextran 4 kDa was then increased, and key pro-inflammatory cytokines, including TNF-alpha and IL-6, were substantially released. Whereas in the M1 macrophage-like THP-1 co-culture model, we could not detect the release of IL-23, which plays a crucial regulatory role in IBD, this cytokine was clearly detected when using primary human M1 macrophages instead. In conclusion, we provide an advanced human in vitro model that could be useful for screening and evaluating therapeutic drugs for IBD treatment, including potential IL-23 inhibitors.

## 1. Introduction

The “leaky gut”, also known as increased intestinal permeability, describes a damaged (leaky) intestinal barrier caused by the loose tight junctions of intestinal epithelial cell walls. This phenomenon results in the passage of harmful substances such as pathogens and toxic digestive metabolites from the gut into the bloodstream, and then consequently causes systemic inflammation and immune system activation. An increased intestinal permeability has been considered to play an important role in the development and progression of numerous chronic diseases [1,2,3]. Autoimmune diseases [4,5], food sensitivities and allergies [6,7], asthma [8], neurological conditions [9,10,11], autism spectrum disorder [12,13], and gut-related disorders like chronic inflammatory bowel disease (IBD) [14,15,16] have been reported in association with a “leaky gut”. IBD is an umbrella term that is used mainly to describe two chronic inflammatory conditions of the gastrointestinal (GI) tract: ulcerative colitis (UC) and Crohn’s disease (CD). In 2017, about seven million people were suffering from IBD worldwide [17], with the prevalence surpassing 0.3% of the general population in North America, Oceania, and many European countries. Moreover, IBD has become a global disease in the twenty-first century with rising incidence and prevalence in many regions, particularly in the emerging economies of South America, Eastern Europe, Asia, and Africa [18]. As there is currently no specific cause and cure for IBD, it presents a tremendous financial burden globally due to the substantial direct costs of medical care and the indirect costs related to disability and missed work [19].

To better understand underlying mechanisms and ultimately identify effective treatment options, many researchers have established various cell-based in vitro models of IBD. For decades, the Caco-2 cell line, a heterogeneous human epithelial colorectal adenocarcinoma cell line, has undoubtedly become the most widely accepted in vitro cell model to study the intestinal absorption of drugs, cell membrane permeability, and inflammatory response [20,21,22,23,24,25,26]. When growing as a confluent monolayer on inserts, Caco-2 cells differentiate and demonstrate morphological and functional characteristics of small intestinal absorptive cells such as tight junctions, and a brush border with well-developed microvilli on the apical surface [27]. Furthermore, Caco-2 cells also express typical enzymes of normal small-intestinal villus cells, such as disaccharidases and peptidases [28].

On the other hand, this single-cell Caco-2 model has been criticized by many authors, because it lacks mucus production. Based on the Caco-2 cell model, many modifications and improvements have followed [29]. One of the most important enhancements was combining it with the mucin-secreting HT29-MTX-E12 goblet cell line [30]. The proportion of goblet cells among epithelial cell types ranges from 10% in the small intestine to 24% in the distal colon [31]. Therefore, the Caco-2/HT29-MTX-E12 co-culture model then better resembled the human small intestine. In the in vitro system, the presence of mucus can act as an interactive barrier, limiting the free diffusion of small compounds to the cells, and thus helping to avoid overestimation of the permeability of such compounds [32]. Moreover, an increasing number of studies reported that the disruption of bidirectional communication between the intestinal mucus barrier and gut microbiota plays a critical role in the development and progression of several inflammatory conditions such as IBD [33,34].

In IBD, intestinal macrophages become activated, and contribute to chronic intestinal inflammation [35,36,37]. Intestinal macrophages in the subepithelial lamina propria (LP) are the most abundant mononuclear phagocytes in the body and play a critical role in maintaining intestinal homeostasis. Thus, some researchers have started to combine intestinal cell lines with human macrophage-like cells (e.g., differentiated monocytic THP-1 cells), or monocyte-derived macrophages from peripheral blood mononuclear cells (PBMCs) [38,39,40]. However, each of these IBD models has its disadvantages, because of either (1) the lack of mucus-producing cells [41,42,43,44], (2) a spatial distance between the intestinal cells and macrophages [45,46,47], or (3) being specifically adapted for buoyant particles [48].

In this study, we developed a complex in vitro triple-culture model of the human intestine, consisting of differentiated human intestinal Caco-2 cells, HT29-MTX-E12 mucus-producing goblet cells (90:10 ratio), cultured on inserts in close contact with either differentiated human macrophage-like THP-1 cells, or primary human monocyte-derived macrophages obtained from PBMCs of healthy donors. The combination of human cell line-derived intestinal cells and macrophages provides a more biological and physiological representation of the complex interactions between the intestinal epithelium and the immune system in health and IBD.

## 2. Results

### 2.1. Establishment of an Inflammation-Triggered, Triple-Culture In Vitro “Leaky Gut” Model Using Caco-2/ HT29-MTX-E12 Co-Culture and Macrophage-like THP-1 Cells

A graphic description of the cell culture and the setup of the inflammation-triggered, triple-culture in vitro “leaky gut” model is demonstrated in Figure 1.

### 2.2. Custom-Designed Three Dimension (3D)-Printed Cap for Medium Confinement in the Insert

A cap was 3D designed (Figure 2A–C) to fit into the insert and sealed the setup by friction. The base of the cap was shaped conically to eliminate possible air pockets when the cap was inserted into the insert (Figure 2D,E).

### 2.3. Characterization of 21-Day-Differentiated Caco-2/HT29-MTX-E12 (90:10 Ratio) Co-Culture

The development of cell monolayer integrity of the Caco-2/HT29-MTX-E12 co-culture during the 21-day incubation was determined by transepithelial/transendothelial electrical resistance (TEER) measurement, which is a widely accepted quantitative technique to assess the integrity of cellular barriers in cell culture models [49]. As presented in Figure 3A, the TEER value of HT29-MTX-E12 cells increased from 18 ± 9 Ω·cm^2^ to 128 ± 18 Ω·cm^2^ between day 3 to 21 of cultivation. In contrast, the TEER values of the Caco-2 monoculture, and the Caco-2/HT29-MTX-E12 co-culture reached 660 ± 31 Ω·cm^2^ and 605 ± 29 Ω·cm^2^ on day 21, respectively. Therefore, in this study, only co-cultures of Caco-2/HT29-MTX-E12 with TEER values above 300 Ω·cm^2^ were used in further experiments.

To determine the presence of mucus on the cell surface, the cell layers of 21-day cultured Caco-2, HT29-MTX-E12, and Caco-2/HT29-MTX-E12 (90:10 ratio) were stained with Alcian blue. Figure 3B shows that the Caco-2 monoculture did not exhibit Alcian blue staining. In contrast, as expected, the surface of HT29-MTX-E12 cells was covered by mucus production, while it was randomly dispersed throughout the cell layer of the Caco-2/HT29-MTX-E12 co-culture.

### 2.4. Increased Intestinal Permeability in Caco-2/HT29-MTX-E12 Co-Culture Induced by IFN-γ Priming

A previous study has reported that IFN-γ priming resulted in the expression of TNF receptor 2, which was crucial for the subsequent induction of TNF-α-induced intestinal epithelial barrier dysfunction, caused by LPS [50]. Therefore, the Caco-2/HT29-MTX-E12 co-culture was primed here with IFN-γ for 24 h, before the addition of the LPS-stimulated macrophage-like THP-1 cells, or primary monocyte-derived macrophages. As shown in Figure 4, IFN-γ priming resulted in a significant drop in the TEER value (Figure 4A) and an increase in permeability (Figure 4B) as signs of barrier integrity loss. Compared to the untreated co-culture (SC), there was a TEER reduction of 25% and a permeability increase of around 18%.

### 2.5. Characterization of the Inflammation-Triggered, Triple-Culture “Leaky Gut” Model

After 6 h and 24 h co-incubation with stimulated macrophage-like THP-1 cells, the intestinal barrier function of the Caco-2/HT29-MTX-E12 epithelial cell layer was assessed by measuring TEER (Figure 5A) and FITC-Dextran 4 kDa (FD4) paracellular transmission (Figure 5B). As seen in Figure 5A, within 6 h, the TEER of the “leaky gut” model decreased to 85% from the initial value, and further to 77% after 24 h. The paracellular flux of FD4 across the cellular layer showed a significantly increased permeability under “leaky gut” conditions, which was up to 41% and further to 71% higher than the control model after 6 h and 24 h, respectively (Figure 5B).

In comparison to the control model, significant release of the cytokines IL-6 and TNF-α, (insignificant for IL-1β), was observed in the “leaky gut” model after 6 h (Figure 5C), and 24 h (Figure 5D). As seen in Figure 5C, after 6 h, the mean concentration of IL-6 and TNF-α in the “leaky gut” model were 66 ± 8 pg/mL and 163 ± 72 pg/mL, respectively. Those were significantly higher than levels of IL-6 and TNF-α in the control model (6 ± 8 pg/mL and 41 ± 22 pg/mL, respectively). However, there was no significant difference in IL-1β production between the “leaky gut” and control models (16 ± 8 pg/mL and 57 ± 50 pg/mL, respectively). Figure 5D also shows the significantly higher levels of IL-6 and TNF-α in the “leaky gut” model (77 ± 26 pg/mL and 91 ± 41 pg/mL, respectively), as compared to the control model (11 ± 4 pg/mL and 30 ± 17 pg/mL, respectively) after 24 h cultivation. Again, there was no significant difference in IL-1β production between the “leaky gut” and control models after 24 h (7 ± 5 pg/mL and 20 ± 13 pg/mL, respectively).

The significant increased permeability highly corresponded to a significant production of zonulin, a proposed regulator of the permeability of the intestinal barrier [51], in the “leaky gut” model after 6 h (Figure 5E) and 24 h (Figure 5F). As seen in Figure 5E, within 6 h, the mean concentration of zonulin in both compartments of the “leaky gut” model (2043 ± 464 pg/mL) was nearly 25-fold higher than in the control model (83 ± 143 pg/mL). After 24 h, the mean concentration of zonulin in both compartments of the “leaky gut” model (1982 ± 260 pg/mL) was around 1.3-fold higher than the control model (1664 ± 29 pg/mL) (Figure 5F).

As shown in Figure 5E,F, while there was no change in Tight Junction Protein-1 (TJP1) between the “leaky gut” and control models after 6 h (24 ± 7 pg/mL and 22 ± 8 pg/mL, respectively) and 24 h (141 ± 12 pg/mL and 147 ± 6 pg/mL, respectively), the amount of released occludin in the “leaky gut” model increased from 66 ± 13 pg/mL at 6 h to 104 ± 49 pg/mL at 24 h. On the other hand, the amount of occludin in the control model decreased from 54 ± 15 pg/mL (6 h) to 51 ± 60 pg/mL (24 h).

### 2.6. Comparison of Different Modifications on the “Leaky Gut” Model Using Macrophage-like THP-1 Cells

A previous study has described that increased intestinal permeability is correlated with increased levels of LPS in intestinal tissue and plasma [52]. Therefore, the “leaky gut” model was additionally treated with 100 ng/mL LPS in the apical compartment for 24 h. As shown in Figure 6, this modification resulted in a significant TEER drop of 30% (Figure 6A) and a permeability increase of 28% (Figure 6B) compared to the control model.

The cytokine interleukin-23 (IL-23), which is primarily produced by macrophages and dendritic cells in response to microbial stimulation, has been considered a key promoter of chronic intestinal inflammation, especially in IBD [53]. However, we could not detect IL-23 secretion in our model (Figure 6C). Lactic acid (LA) was reported as a stimulator of IL-23 production in PBMCs exposed to bacterial LPS [54]. Therefore, we pre-treated macrophage-like THP-1 cells with 1 mg/mL LA for 24 h before setting up the triple-culture. This modification showed a significant TEER reduction of approximately 24% (Figure 6A) and a permeability rise of 13% (Figure 6B) compared to the control model. The combination of the two modifications (apical treatment with LPS and LA priming) also resulted in a significant TEER decrease of 20%, and a permeability increase of 27% as compared to the control model.

As described in Figure 6C, similar to the inflamed model, a significant secretion of IL-6 was found after additional modifications by using apical treatment of 100 ng/mL LPS (66 ± 37 pg/mL), or 24 h-priming of macrophage-like THP-1 cells with 1 mg/mL LA (79 ± 35 pg/mL), or the combination of these (83 ± 37 pg/mL) as compared to the control model (11 ± 4 pg/mL). There were similar levels in the production of IL-1β, TNF-α, and especially IL-23 between the modified “leaky gut” and control models.

### 2.7. Comparison of Different Modifications on the “Leaky Gut” Model Using Primary Human-Derived Macrophages

To better reflect the properties of primary macrophages in vivo, we replaced the THP-1 cell line in our model with primary monocyte-derived macrophages from human PBMCs. The experiments were then carried out as with the macrophage-like THP-1 cells.

The additional treatment with 100 ng/mL LPS in the apical compartment of the “leaky gut” model resulted in a significant TEER drop of 24% (Figure 7A) and a permeability increase of approximately 30% (Figure 7B) as compared to the control model.

Similarly, 1 mg/mL LA pre-treatment of primary monocyte-derived macrophages for 24 h also caused a significant TEER reduction of 25% (Figure 7A), and a permeability rise of 20% (Figure 7B) in comparison with the control model. The combination of the two above modifications (apical treatment with LPS and LA priming) also resulted in a significant TEER decrease of 22%, but the permeability only slightly increased by 8% as compared to the control model.

As shown in Figure 7C, additional modifications with the apical treatment of 100 ng/mL LPS (433 ± 102 pg/mL), or 24 h-priming of macrophages with 1 mg/mL LA (290 ± 130 pg/mL), or the combination of these two modifications (451 ± 80 pg/mL) all produced a significant secretion of IL-6 in comparison with the control model (58 ± 70 pg/mL). Similarly, the substantial secretion in TNF-α was also seen by the apical treatment of 100 ng/mL LPS (741 ± 223 pg/mL), 24 h-priming macrophage with 1 mg/mL LA (662 ± 240 pg/mL), and the combination of these two modifications (1163 ± 314 pg/mL) as compared to the control model (34 ± 21 pg/mL). In contrast to macrophage-like THP-1 cells, the apical treatment with 100 ng/mL LPS (184 ± 180 pg/mL), or 24 h-priming of macrophages with 1 mg/mL LA (209 ± 65 pg/mL), or the combination of these two modifications (577 ± 161 pg/mL) resulted in a strong, significant secretion of IL-23 when compared to the control model (0 pg/mL).

## 3. Discussion

IBD is a chronic gastrointestinal inflammatory disease with unclear causes and pathogenesis. However, it is thought that a complex sequence of interactions among genetic, microbial, immunological, and environmental factors results in an abnormal and exaggerated immune response of the commensal microbiota, finally resulting in the induction of intestinal inflammation [55,56,57,58].

Various in vitro models have been developed to understand IBD’s etiology, pathology, and potential treatment options. Most of them used the Caco-2 cell line, because differentiated Caco-2 cells can reflect many features of mature enterocytes in the intestinal epithelium, such as a brush border with microvilli, tight junction formation, production of characteristic digestive enzymes, and transporters [27]. Importantly, Caco-2 cells showed the ability to produce a range of inflammatory cytokines such as IL-6, IL-8, IL-1β, and TNF-α, that may contribute to inflammatory conditions in IBD [59]. Different studies have also utilized this cell line to evaluate novel molecules for potential therapeutic treatment/management of UC. For example, it was demonstrated, that rhamnogalacturonan accelerated wound healing, decreased epithelial barrier dysfunction, and suppressed IL-1 induced IL-8 production in Caco-2 cells [60]. Another study by Liang et al. reported that the corn protein hydrolysate down-regulated the secretion of IL-8 production in TNF-α induced inflammation in Caco-2 cells [61].

Nevertheless, the most significant limitation of the monolayer Caco-2 cell culture system was the lack of a mucus layer, which serves as a physical and chemical barrier in the intestinal epithelium against luminal contents involving digestive enzymes, food particles, microbiota and microbial compounds, as well as host-secreted products such as bile acids [62]. Hence, previous studies co-cultured Caco-2 cells with the mucus-secreting goblet HT29-MTX-E12 cell line to provide a closer physiological model of the human intestinal epithelium [63,64,65]. Many recent studies have widely used the optimal 90:10 ratio of Caco-2/HT29-MTX cells, because it better represented the in vivo situation of the human small intestine regarding the proportion of absorptive enterocytes and mucin-producing goblet cells [66,67,68,69]. Therefore, we applied this combination of Caco-2/HT29-MTX-E12 co-culture conditions for our “leaky gut” model system to closely mimic the cell composition and functionality of the intestinal epithelium. In our model, the mucin secreted by the HT29-MTX cells does not appear to completely cover the whole surface of the intestinal cells after 21 days of co-culture. A healthy mucosal barrier contributes to the prevention of pathogens invasion and defects therein have been implicated in several intestinal pathologies. Thus, depending on the application of our in vitro model, this might be considered as a limitation or advantage.

Macrophages in the lamina propria of the small intestine are one of the most prevalent populations of leukocytes in the body. They play a crucial role in maintaining intestinal homeostasis and intestinal inflammation emergence [70]. Several investigations have demonstrated that intestinal macrophages become activated and promote the occurrence and development of IBD [35,37,71]. Over the last decade, to better replicate the in vivo physiology of IBD, in vitro IBD models have been established using a combination of Caco-2 cells and macrophage-like differentiated THP-1 cells, for example, Kämpfer et al. (2017) [46]. IFN-γ priming of Caco-2 cells together with the stimulation of differentiated THP-1 cells by LPS and IFN-γ, induced an inflammation-like response in their diseased intestine model, evident by intestinal barrier disruption and pro-inflammatory cytokine release. Based on this model of Kämpfer, co-culture models with differentiated Caco-2 cells and PMA-differentiated THP-1 cells in the presence of inflammatory stimulators (e.g., LPS with/without IFN-γ) have been used to evaluate the potential immunomodulatory and anti-inflammatory effects of phytochemicals [43,72,73], marine natural products [74,75,76], bacterial β-glucans [77], siRNA-based nanomedicine [42], bovine milk-derived extracellular vesicles [78], and probiotic bacteria [41].

Some IBD studies used an advanced in vitro triple-culture model composed of Caco-2 cells, HT29-MTX-(E12) cells, and PMA-differentiated macrophage-like THP-1 cells. The combination of two intestine cell lines (Caco-2/HT29-MTX) with differentiated THP-1 cells was first introduced by Kaulmann et al. (2016) to study the anti-inflammatory and antioxidant effects of plums and cabbages [79]. Busch et al. (2021) suggested that this advanced in vitro triple-culture model was a promising approach for studying the toxicological effects of ingested micro- and nano-plastic particles [80]. An adverse and pro-inflammatory role of the NLRP3 inflammasome in IBD has been described by using this triple-culture model [81].

However, this model system cannot reflect the anatomical distribution of lamina propria macrophages in humans, which are close to the epithelial monolayer of intestinal cells [70]. Thus, Calatayud et al. co-cultured Caco-2, HT29-MTX, and differentiated THP-1 cells in close contact [38]. In contrast to our model, they used Type I collagen from a rat tail to support THP-1 adhering to the membrane. Even though the presence of coated collagen could increase cell attachment and viability [82], it may potentially impact TEER measurements indirectly because the phenotyping properties of THP-1 cells were modified by the surrounding extracellular matrix (i.e., collagen Type I). Teplicky et al. have demonstrated that cell doubling times (i.e., cell proliferation) and mean diameters (i.e., cell size) of THP-1 cells in collagen Type I were slightly increased when compared to cells cultured in normal medium [83]. Furthermore, the biological activity of collagen-coated immune cells and the detection of released pro-inflammatory cytokines might also be affected [84]. Another study by Busch et al. (2021) also used a triple-culture model with close contact between intestinal cells (Caco-2/HT29-MTX-E12) and differentiated macrophage-like THP-1 cells [48]. They seeded Caco-2/HT29-MTX-E12 cells on the bottom side of the insert, while differentiated macrophage-like THP-1 cells were cultured on the top side of the insert. Due to the experimental design of their model, it is only suitable for studies with buoyant particles, which float in cell culture media due to their density of less than 1 g/cm^3^.

A significant increase in zonulin production could be observed in our “leaky gut” model when compared to the control model. Intestinal epithelial cell tight junctions are a multi-protein complex that support the integrity of the physical intestinal barrier by regulating the paracellular movement between the internal environment and external antigens or bacterial products [85,86]. It has been demonstrated that impaired tight junction proteins present an early event of IBD [87,88]. In fact, elevated levels of zonulin have been detected in both serum [89,90,91] and fecal [92,93] samples of IBD patients and it is used as biomarker of intestinal permeability of the small intestine [94,95]. Therefore, on this point our approach of a “leaky gut” model provides in vitro to in vivo concordance.

Cytokines play a critical role in the immunopathogenesis of IBD, where they regulate various aspects of the inflammatory response [96,97,98]. In patients with IBD, pro- and anti-inflammatory cytokines have been demonstrated to be produced in the inflamed mucosa by various immune cells such as macrophages, dendritic cells (DCs), neutrophils, natural killer (NK) cells, intestinal epithelial cells (IECs), innate lymphoid cells (ILCs), mucosal effector T cells (T helper 1 (T_H_1), T_H_2 and T_H_17), and regulatory T (T_reg_) cells [99]. In particular, the translocation of commensal bacteria and microbial products from the gut lumen into the bowel wall resulting from an impaired cell barrier function (“leaky gut”) leads to inflammatory macrophage (M1 phenotype) stimulation, and consequent production of high levels of pro-inflammatory cytokines such as IL-1, IL-6, IL18, TNF-α, IL-23, and IL-17. These cytokines directly or indirectly result in the injury or necrosis of the intestinal epithelial cells, which then promotes the pathogenesis of IBD [35]. Being critical mediators in the development of IBD, pro-inflammatory cytokines are considered effective therapeutic targets [100,101]. Anti-TNF-α therapy is the first biologic approved, and currently the most effective treatment for IBD, including infliximab, golimumab, adalimumab, and certolizumab pegol, which have been demonstrated good clinical efficacy [102]. However, approximately 20% of IBD patients are primary non-responders [103], and over 30% eventually lose response to anti-TNF drugs [104]. Blocking of lamina propria macrophages-produced IL-6 with monoclonal antibodies (e.g., tocilizumab, PF-0423691) is here considered as alternative treatment for IBD, but serious side effects have been reported for these anti-IL-6 drugs [32].

More recent data have demonstrated, that the pro-inflammatory IL-23 was a critical promoter of the pathogenesis of IBD, because it stimulates and influences the differentiation and proliferation of pathogenic T helper type 17 (T_h_17) cells. This in turn further induces inflammatory cytokines [53,105,106]. Therefore, targeting the IL-23 pathway is another important way for drug development of IBD [107]. Currently, only ustekinumab has been approved for the treatment of both CD and UC patients, but several IL-23p19 antagonists (e.g., risankizumab, brazikumab, mirikizumab) are in phase II or III development programs and give promising results. Our “leaky gut” model showed a significant increase in IL-6 and TNF-α upon activation. Interestingly, we could also achieve the substantial secretion of IL-23 by additional modifications using primary human-derived macrophages, but not by using macrophage-like THP-1 cells. Since the exact mechanism by which lactic acid stimulates macrophages to release IL-23 is not fully understood, one possible reason for this difference could be correlated to the genetic and phenotypic differences between macrophage-like THP-1 cells, and primary blood macrophages. Compared to primary blood macrophages belonging to a non-malignant and non-proliferating cell type, THP-1 cells are leukemia monocytic cells with genetic and functional differences. Furthermore, concerning LPS responses, THP-1 cells express much lower levels of monocyte differentiation antigen CD14 in comparison to primary monocytes [108]. The detection of IL-23 has not yet been described in any other intestinal inflamed model before, thus our “leaky gut” model provides a promising new in vitro platform for drug investigation of IBD treatment, especially IL-23 pathway inhibitors.

## 4. Materials and Methods

### 4.1. Chemicals

Fetal calf serum (FCS), GlutaMAX^TM^ supplement, Roswell Park Memorial Institute (RPMI)-1640, Dulbecco’s Modified Eagle’s medium (DMEM) with low glucose, trypsin-EDTA (0.5%), trypsin (2.5%) solution, phosphate-buffered saline (PBS, without Ca^2+^ and Mg^2+^), Non-Essential Amino Acid (NEAA), penicillin/streptomycin solution (10,000 U/mL and 10,000 µg/mL), StemPro^TM^ Accutase^TM^, and Hank’s Balanced Salt Solution (HBSS) were purchased from Gibco™, Life Technologies GmbH (Darmstadt, Germany). Lipopolysaccharide (LPS, from Escherichia coli O111:B4), phorbol 12-myristate 13-acetate (PMA), fluorescein isothiocyanate (FITC)-Dextran, Alcian blue 8GX solution (1% in 3% acetic acid) were from Sigma Aldrich (Taufkirchen, Germany). IFN-γ (human recombinant) was purchased from STEMCELL technologies GmbH (Köln, Germany). Macrophage colony-stimulating factor (M-CSF) was purchased from Peprotech (Hamburg, Germany). Paraformaldehyde (PFA) solution of 4% in PBS was purchased from Santa Cruz Biotechnology (Heidelberg, Germany).

### 4.2. Cell Culture

The human colon carcinoma Caco-2 (ACC169) and HT29-MTX-E12-E12 cell lines were obtained from the German Collection of Microorganisms and Cell Cultures (DSMZ, Braunschweig, Germany) and European Collection of Authenticated Cell Cultures (Porton Down, UK), respectively. The cells were cultured separately in flasks in DMEM supplemented with 10% (*v*/*v*) FCS, 1% (*v*/*v*) NEAA, 100 U/mL penicillin, and 100 µg/mL streptomycin at 37 °C in a humidified incubator with a 5% CO_2_/95% air atmosphere. The culture medium was changed every 2–3 days and cells were regularly split at 90% confluence.

The THP-1 cell line was cultured in a flask in RPMI-1640 supplemented with 10% (*v*/*v*) FCS, 1% GlutaMAX^TM^, 100 U/mL penicillin, and 100 µg/mL streptomycin at 37 °C in a humidified incubator with a 5% CO_2_/95% air atmosphere. THP-1 cells were maintained at a concentration between 0.2 to 1 × 10^6^ cells/mL.

### 4.3. Isolation and Cultivation of Human PBMCs

Human PBMCs were isolated from buffy coats of healthy volunteers at the University Medical Center in Freiburg, Germany by centrifugation on a LymphoPrep^TM^ gradient (density: 1.077 g/cm^3^, 20 min, 500× *g*). Isolated PBMCs were cultured in complete RPMI 1640 medium supplemented with 10% heat-inactivated FCS, 2 mM L-glutamine, 100 U/mL penicillin, and 100 µg/mL streptomycin at 37 °C in a humidified incubator with a 5% CO_2_/95% air atmosphere.

### 4.4. Co-Culture of Caco-2 and HT29-MTX-E12-E12 on Inserts

Monocultures of Caco-2 and HT29-MTX-E12 cells were harvested with Trypsin-EDTA and seeded on the apical chamber side of 12-well ThinCert^®^ inserts (0.4 µm PET pore membrane, Greiner Bio-One, Frickenhausen, Germany) in an optimal proportion of 90:10, respectively, to reach a final density of 1 × 10^5^ cells/cm^2^/insert. Cells were co-cultured for 19–21 days in a humidified incubator with a 5% CO_2_/95% air atmosphere with medium (0.5 mL on the apical side and 1.5 mL on the basolateral side) changed every 2–3 days.

### 4.5. Macrophage Differentiation from THP-1 Cell Line and Peripheral Blood Primary Monocytes

THP-1 monocytes were seeded at 2 × 10^5^ cells/mL in a 75 cm^2^ flask and differentiated into macrophages by 72 h treated with 20 ng/mL PMA in a 5% CO_2_/95% air atmosphere incubator at 37 °C. After differentiation, the PMA-containing medium was discarded and the macrophage-like differentiated THP-1 were rested in fresh medium for 24 h.

Primary monocytes were purified from isolated PBMCs by using the culture plastic adherence technique: 2 × 10^6^ isolated PBMCs/mL in complete medium were seeded into a 75 cm^2^ culture flask and then monocytes were allowed to adhere at 37 °C in a 5% CO_2_/95% air atmosphere incubator. After 24 h incubation, non-adherent cells were removed from the flask. For macrophage differentiation, the adherent cells (mainly monocytes) were fed with the complete medium containing 50 ng/mL recombinant human M-CSF for additional 6 days in a 5% CO_2_/95% air atmosphere at 37 °C. The medium was then replaced every 3 days with fresh complete medium, supplemented with 50 ng/mL M-CSF. After that, monocyte-derived macrophages were rested in fresh medium for 24 h.

### 4.6. Fabrication of 3D-Printed Cap for Insert

The caps were fabricated with a FormLabs 3+ 3D printer and BioMed Clear Resin, a USP class VI material used for medical devices that complies with ISO 18562. To avoid possible cross-contamination of the used materials, the caps were fabricated in an ISO 13485-certified laboratory with a printer that was solely used for the respective material. After printing, the caps were washed (Form Wash) and cured (Form Cure) according to the manufacturer’s instructions (all devices and materials were purchased from FormLabs GmbH, Berlin, Germany). The Appendix A of the cap can be downloaded as Appendix A from the journal’s homepage.

### 4.7. Experimental Setup of the Inflammation-Triggered “Leaky Gut” Model

The control and inflammation-triggered or “leaky gut” Caco-2/HT29-MTX-E12-E12/THP-1 triple-culture was established as illustrated in Figure 1. Firstly, 19-day-differentiated Caco-2/HT29-MTX-E12-E12 epithelial cells in the apical compartment were either rested in fresh medium (control model) or primed with 50 ng/mL IFN-γ for the inflammation-triggered model in 24 h before the triple-culture. On the next day, the medium from the basolateral side of the insert was completely discarded. The medium from the apical part was replaced with fresh medium, and a specially 3D-printed constructed cap was carefully placed into the insert (see Figure 2) to avoid the leakage of the medium during the following immune cell adherence procedure. The insert was placed upside down in a Petri dish. Then THP-1-differentiated macrophages (2 × 10^5^ cells), or primary monocyte-derived macrophages (4 × 10^4^ cells), which were initially detached from the 75 cm^2^ culture flask with 50 ng/mL accutase, were transferred on the bottom side of the inverted inserts, for 1.5 h at 37 °C in a 5% CO_2_ incubator. After immune cell adherence, the inserts were put back into 12-well plates in regular orientation before the specially 3D-printed constructed caps were carefully removed from the inserts. After medium removal, fresh medium was added to the upper and lower compartments of the insert. Macrophages from the control triple-culture were rested in fresh medium, while macrophages from the inflammation-triggered triple-culture were activated by 100 ng/mL LPS in combination with 10 ng/mL IFN-γ for 24 h at 37 °C in a humidified incubator with a 5% CO_2_/95% air atmosphere.

To further optimize the “leaky gut” model, the treatment procedures involved some additional modifications. Condition 1: macrophage-like THP-1 cells were primed with 1 mg/mL lactic acid (LA) in 24 h before triple culture. Condition 2: the apical compartment was simultaneously treated with 100 ng/mL LPS during macrophage activation in the basolateral compartment. Condition 3: combination of conditions 1 and 2.

### 4.8. Transepithelial Electrical Resistance Measurement

The cell monolayer integrity of the Caco-2/HT29-MTX-E12 co-culture on an insert was investigated using transepithelial electrical resistance (TEER) measurement. This measurement was performed by using an EVOM epithelial volt-ohmmeter equipped with a ‘chopstick’ electrode (STX-2) (Millicell^®^ ERS, Millipore, Bedford, MA, USA). Before measurement, cells were stabilized at room temperature, while the electrode was sterilized with 70% ethanol and preconditioned in growth media. The measurement was performed in triplicates, and immediately after medium replacement. The final TEER value (TEER _final_) was corrected by subtracting the blank resistance (R _blank_) of the semipermeable membrane only (an insert without cells) from the resistance across the sample (R _sample_) before multiplying it by the effective growth area (A) of the insert.
TEER _final_ [Ω × cm^2^] = (R _sample_ − R _blank_) [Ω] × A [cm^2^]

Co-culture inserts with TEER values over 300 Ω·cm^2^ were used for further experiments. TEER results were expressed as a percentage of the initial TEER value.

### 4.9. Alcian Blue Staining

Alcian Blue stain was used to visualize acidic epithelial and connective tissue mucins that were produced by HT29-MTX-E12 cells in the co-culture model. Briefly, Caco-2/HT29-MTX-E12 co-culture (90:10 ratio) was cultured at a density of 1 × 10^5^ cells/cm^2^/insert on 12-well plates for 21 days. After 21-day incubation, culture media were removed and cells were washed twice with pre-warmed PBS before they were fixed with 4% paraformaldehyde (PFA) for 30 min at room temperature. Next, PFA was aspirated and cells were rinsed with PBS twice before they were stained with 1% alcian blue in 3% acetic acid. After 30 min incubation at room temperature, extra alcian blue was removed by two-time washing with PBS. The stained mucus was visualized by an inverted microscope (Fluorescence microscope Biozero BZ 8100E, Keyence GmbH, Neu-Isenburg, Germany).

### 4.10. Permeability Studies

Paracellular permeability of the intestinal epithelium layer was determined using FITC-Dextran with a molecular weight of 4 kDa (FD4). Briefly, 250 µL FD4 solution (1 mg/mL in HBSS), and 800 µL HBSS was added to the apical and basolateral compartment, respectively. After 2 h incubation at 37 °C, 150 µL from the basolateral side were transferred to a black 96-well plate (Greiner Bio-One, Frickenhausen, Germany). HBSS and FD4 solution were used as a negative and positive control, respectively. Fluorescence intensity was measured at excitation and emission wavelengths of 490 and 520 nm, respectively, by using a plate reader (TECAN infinite M200, Tecan tranding AG, Männedorf, Switzerland). Permeability coefficient (*P_app_*) was calculated by using the following equation:$$P_{app}=\frac{dQ}{dt}\times \frac{1}{A\times C_{0}}$$

Defined as:

*P_app_* = apparent permeability coefficient [cm/s].

*dQ*/*dt* = rate of appearance of FD4 on the basolateral side [µg/s].

*A* = surface area of the monolayer [cm^2^].

C_0_ = initial FD4 concentration in the apical side [µg/mL].

### 4.11. Tight Junction Proteins and Their Regulator Quantification

Cells were stimulated as described above. Secreted tight junction proteins (Occludin, Tight Junction Protein 1) and their regulator (Zonulin) in the supernatant were quantified by using specific ELISA kits (AssayGenie, Dublin, Ireland) according to the manufacturer’s instructions. Results were standardized by comparison with a standard curve.

### 4.12. Cytokine Level Measurement

Cells were stimulated as described above. Secreted proinflammatory cytokines (IL-1β, IL-6, IL-23, and TNF-α) in the supernatant of the lower compartment were evaluated by using specific ELISA kits (Thermo Scientific, Darmstadt, Germany) according to the manufacturer’s instructions. Results were standardized by comparison with a standard curve.

### 4.13. Statistical Analysis

Results are expressed as the means ± standard deviation (SD) of at least three independent experiments. When comparing between two groups, Student’s unpaired *t* test was used. For experiments involving more than three groups, results were analyzed either by two-way ANOVA or two-way ANOVA followed by Tukey’s multiple comparison tests. Data were analyzed using the GraphPad Prism version 6.07 software (GraphPad Software Inc., San Diego, CA, USA). Results were considered statistically significant when *p* < 0.05.

## 5. Conclusions

In conclusion, we described the establishment of a complex “leaky gut” model using epithelial cells (i.e., Caco-2), and mucus-secreting cells (i.e., HT29-MTX-E12) in close contact with activated immune cells (i.e., differentiated macrophage-like THP-1 cells or primary monocyte-derived macrophage) to simulate pathophysiological mechanisms of intestinal inflammation. Modifications on the original “leaky gut” model using primary human-derived macrophages, with either the additionally apical LPS treatment, or the LA pre-treatment of macrophages, further increased at least one of the “leaky gut” characteristics in our model. In particular, the expression of IL-23 could present a further advantage of this in vitro model. Even though there is no single model that can mimic all complex aspects of IBD, we could address some limitations of previously established models. Therefore, our in vitro “leaky gut” model can provide a promising pre-clinical tool for novel IBD-related drug development and serve as an alternative system to in vivo animal testing.

## Figures and Tables

**Figure 1 ijms-24-07427-f001:**
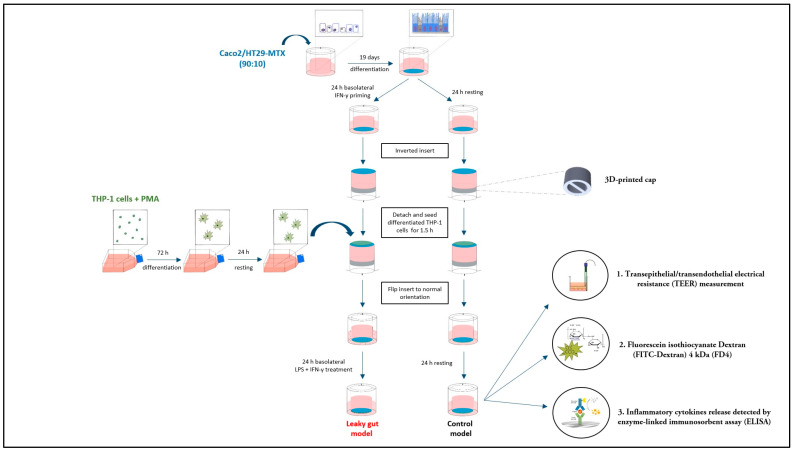
Schematic diagram of the inflammation-triggered, triple-culture in vitro “leaky gut” model. 19-day-differentiated Caco-2/HT29-MTX-E12 co-cultures were either rested in fresh medium (control), or primed for 24 h with IFN-γ. On the following day, a custom-designed three dimension (3D)-printed cap was carefully placed into the insert of the cultures to confine the medium, before placing the insert upside down in a Petri dish. After that, phorbol 12-myristate 13-acetate (PMA)-differentiated macrophage-like THP-1 cells, or primary monocyte-derived macrophages were transferred on the bottom side of the inverted inserts for 1.5 h, before flipping it back to the regular orientation. Then, for generating the inflammation-mediated “leaky gut” condition, macrophages were activated for 24 h by adding a combination of LPS and IFN-γ. At the same time, for the control model, macrophages were rested in medium for this time.

**Figure 2 ijms-24-07427-f002:**
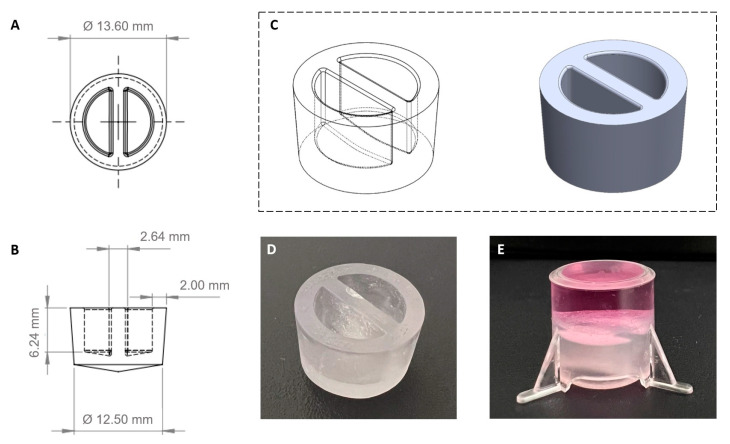
Custom-designed 3D-printed cap. The design sketches of the cap are: top view (**A**), front view (**B**), and 3D view (**C**). The dimensions of this cap (**D**) were determined to fit into the insert perfectly to safely confine the medium within the insert during the macrophage adherence procedure (**E**).

**Figure 3 ijms-24-07427-f003:**
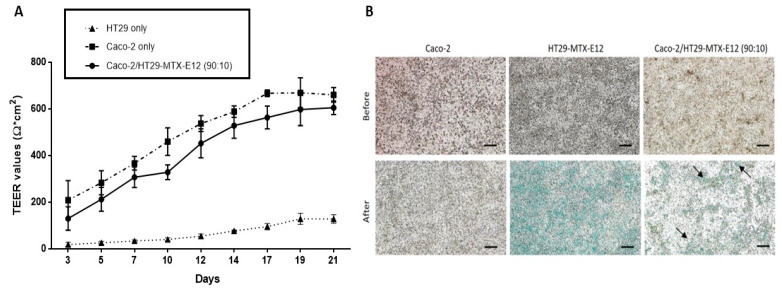
Characterization of the Caco-2/HT29-MTX-E12 (90:10 ratio) co-culture during 21 days of cultivation. (**A**) Transepithelial-transendothelial electrical resistance (TEER) values of Caco-2, HT29-MTX-E12, and Caco-2/HT29-MTX-E12 (90:10) cell cultures. Values are shown as mean ± standard deviation (SD) (*n* ≥ 3). (**B**) Representative microscopy images of cell layers were obtained before and after staining with Alcian blue for mucus production (evident by blue color and black arrows). Scale bar = 100 µm.

**Figure 4 ijms-24-07427-f004:**
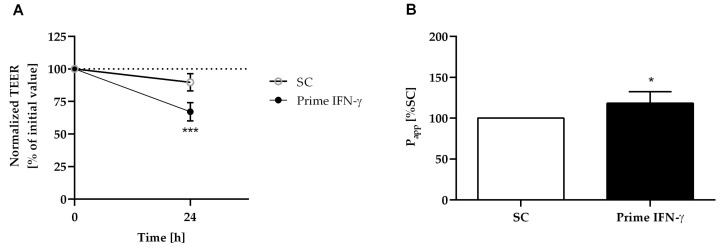
IFN-γ priming on the Caco-2/HT29-MTX-E12 co-culture. After 24 h incubation with 50 ng/mL IFN-γ, TEER (**A**) and permeability value (P_app_) (**B**) of Caco-2/HT29-MTX-E12 co-cultures were measured. TEER values were expressed as a percentage of the initial TEER value (100%). Permeability was expressed as fold change of the untreated control (SC) used as a reference. Bars are the means ± SD (*n* ≥ 3). * *p* < 0.05, or *** *p* < 0.001 were considered significantly different versus SC.

**Figure 5 ijms-24-07427-f005:**
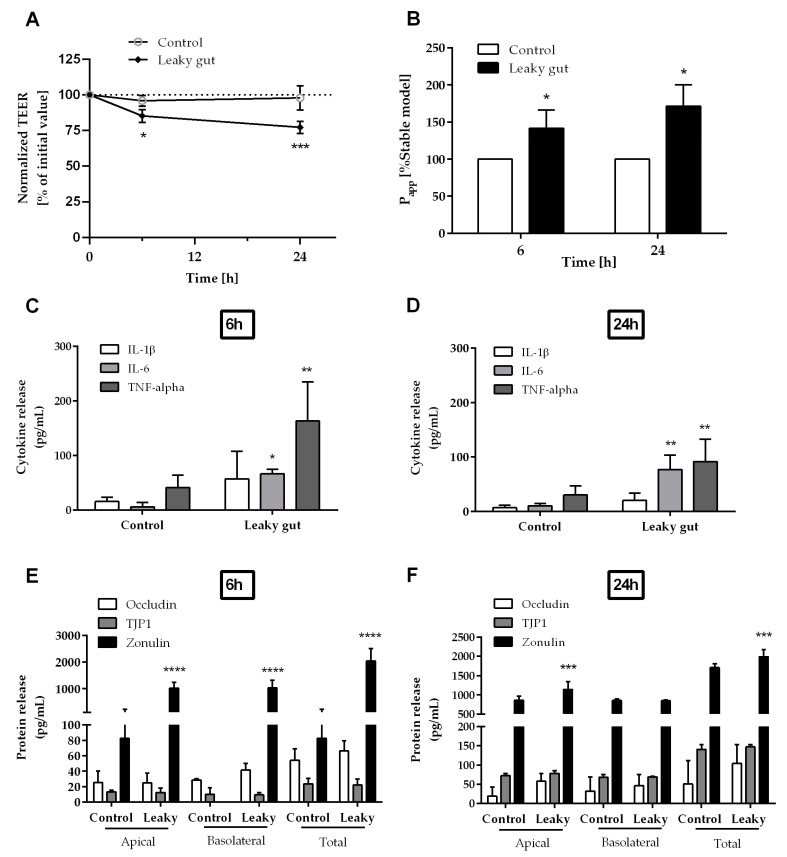
Characterization of the in vitro “leaky gut” model. Intestinal barrier function of the triple cell culture was assessed by (**A**) TEER and (**B**) permeability value (P_app_) after 6 h and 24 h incubation. The secretion of the key pro-inflammatory cytokines IL-1β, IL-6, and TNF-α in the basolateral compartment were quantified after (**C**) 6 h and (**D**) 24 h. The release of Occludin, Tight Junction Protein-1 (TJP1), and Zonulin in the apical, and basolateral compartment were measured after (**E**) 6 h and (**F**) 24 h. Data are given as mean ± SD (*n* ≥ 3). * *p* < 0.05, ** *p* < 0.01, *** *p* < 0.001, **** *p* < 0.0001 as compared to the control model.

**Figure 6 ijms-24-07427-f006:**
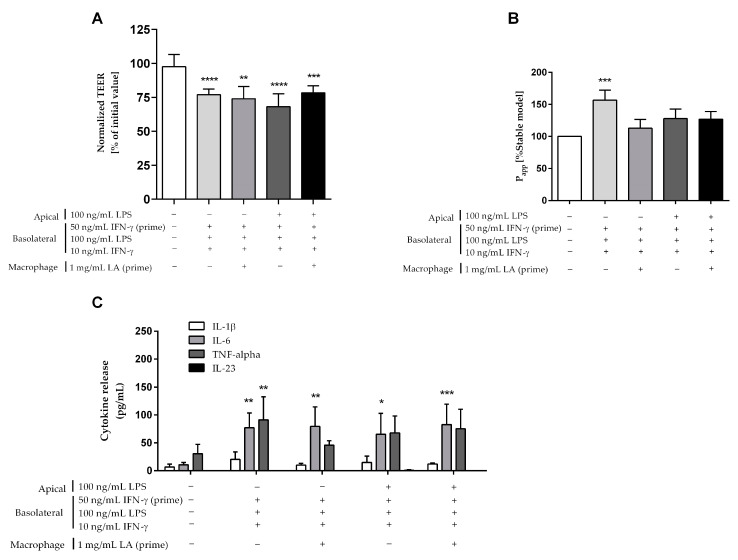
Comparison of different modifications on the “leaky gut” model using macrophage-like THP-1 cells (“−“: non-treatment, “+”: treatment). Intestinal barrier function was assessed by (**A**) TEER and (**B**) permeability value (P_app_) as compared to the control model. (**C**) The secretion of key pro-inflammatory cytokines IL-1β, IL-6, IL-23, and TNF-α was quantified in the basolateral compartment of the models after 24 h cultivation. Bars are mean ± SD (*n* ≥ 3). * *p* < 0.05, ** *p* < 0.01, *** *p* < 0.001, **** *p* < 0.0001 as compared to the control model.

**Figure 7 ijms-24-07427-f007:**
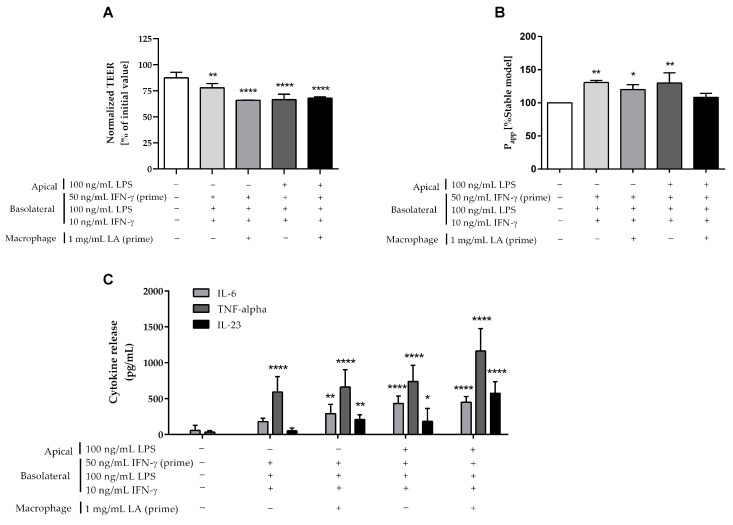
Comparison of the modifications on the “leaky gut” model using primary human-derived macrophages (“−”: non-treatment, “+”: treatment). Intestinal barrier function in comparison to the control model was assessed after 24 h by (**A**) TEER and (**B**) permeability values. (**C**) The secretion of key pro-inflammatory cytokines IL-6, IL-23, and TNF-α in the basolateral compartment of control and “leaky gut” models were quantified after 24 h. Bars are mean ± SD (*n* ≥ 3). * *p* < 0.05, ** *p* < 0.01, **** *p* < 0.0001 as compared to the control model.

## Data Availability

Not applicable.

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
