# Peer review of "Development of an Inflammation-Triggered In Vitro “Leaky Gut” Model Using Caco-2/HT29-MTX-E12 Combined with Macrophage-like THP-1 Cells or Primary Human-Derived Macrophages"

_ijms, 2023, doi:10.3390/ijms24087427_

Round 1

Reviewer 1 Report

In vitro and ex vivo IBD models are of interest to many research groups due to the high incidence of inflammatory bowel diseases. Among the many tested cell lines, the most commonly used are Caco-2 and HT-29, which are currently the gold standard. Both are human colon-cancer-derived cell lines. They are particularly useful because of their ability to express characteristics of mature intestinal cells such as enterocytes or mucus generating cells. BD models require the participation of immune cells that are actively involved in the development of inflammation. The most commonly used are PBMC, THP-1 (human origin) or RAW.246.7 (murine origin). Many variants of in vitro models have been described, even if their authors used the same cell lines. The advantages and disadvantages of different versions of co-culturing have already been described in detail. The model used in this work is considered stable and reproducible, very good for research of bacterial adhesion, colonic permeability, and appropriate for studying immune related mechanisms. It is important here to determine the extent of changes in the interaction between immune cells and the intestinal cells (appropriately differentiated cell line cells). I wonder how much trouble researchers had with maintaining cell viability in such a complex system. There is currently a competition to develop the most appropriate models to study the interaction between innate immune cells and model gut cells. It is important to choose the right source of innate immune cells with high plasticity that allows their functional differentiation, imitating physiological conditions as closely as possible. I congratulate the Authors of the work for going beyond simple 2D cultures. They developed a triple-culture model consisting of differentiated Caco-2 cells, HT29-MTX-E12 mucus-producing goblet cells and differentiated human macrophage-like THP-1 cells, or primary human monocyte-derived macrophages obtained from PBMCs. Technical details are described in great detail with the advantages and disadvantages noted. Good work for researchers looking for mechanisms and potential therapies! In my opinion, the manuscript can be accepted without any critical comments.

Author Response

Dear Reviewer,

Sincerely yours,

Nguyen Le

Reviewer 2 Report

The combination of Caco-2/HT29-MTX-E12 with macrophage-like THP-1 cells or primary human-derived macrophages is intended to create an in vitro leaky gut model. The model more closely resembles living conditions than conventional models. This might be an interesting topic for publication in this journal. The work is, for the most part, sound, and the experiments are well designed and executed. I have comments, explained below. I hope that my comments are very useful for the improvement of this research.

Comments

(1) Title: I think the title does not correctly reflect the experimental data. I believe that the expression of IL-23 does not occur when THP-1 is used. The only model in which IL-23 is expressed is when primary human-derived macrophages are used.

(2) leaky gut” model: The authors use TEER and FITC-dextran to evaluate intestinal permeability in the leaky gut model. What level of reduction in TEER values or FITC-dextran permeability is sufficient to determine that the model is able to mimic a living organism? Is there a target value for this? This is a very difficult point to consider.

(3) Section 2.3: It is not known if the cell surface mucosa is stained. This is because Alcian blue staining stains the goblet cells as well as the mucin layer. Therefore, the wording of this section needs to be reworded. I recognize that the results indicate that Caco-2 and HT-29 cells can be co-cultured.

(4) Section 2.3: In in vivo, the mucin layer covers the intestinal epithelial cells. In Figure 3B, the mucin layer does not appear to cover the cell surface. Therefore, the mucin layer in this experiment does not mimic a living organism. It is necessary to indicate this point as “Limitation” in the discussion section.

(5) Occludin and TJP1 in the medium in Fig. 5: The authors measure occludin and TJP1 in the medium. To what extent do the concentrations of occludin and TJP1 in the medium correlate with the expression of tight junction proteins in the cells? This point is unclear, please explain with references.

(6) Fig. 5, 6, 7: ml-> mL

Author Response

(The authors gave the same response as above.)

Reviewer 3 Report

The research under the title “Development of an inflammation-triggered, IL-23 expressing in vitro “leaky gut” model using Caco-2/HT29-MTX-E12 combined with macrophage-like THP-1 cells or primary human-derived macrophages” by Le and coworkers presents results on the development of the new model of the “leaky gut.” The conclusions are well-explained and supported by the obtained data from various studies at the molecular level. The article might interest the International Journal of Molecular Sciences readers after the following corrections are made. My recommendation is a Major Revision.

The Authors should address the following:

1.      The authors should mention how the zonulin concentration was determined in the Results section.

2.      Section 2.2. doesn’t have any text; it is only the figure.

3.      Were the changes in occluding concentrations significant?

4.      How was the concentration of LPS determined in section 2.6? Is the amount of TEER drop correlated with the concentration of LPS?

5.      Is the additional change in TEER concentration dependent?

6.      How would the model react to the use of approved antagonists?

7.      What is the Conclusion on the use of two modifications? Is there any synergistic effect?

8.      More the quantitative data should be added to the Conclusion

Author Response

(The authors gave the same response as above.)

Round 2

Reviewer 3 Report

The authors have answered all of the queries by the Reviewer. The article is suitable for publication.